# Human and Mouse Bone Marrow CD45^+^ Erythroid Cells Have a Constitutive Expression of Antibacterial Immune Response Signature Genes

**DOI:** 10.3390/biomedicines13051218

**Published:** 2025-05-17

**Authors:** Roman Perik-Zavodskii, Olga Perik-Zavodskaia, Julia Shevchenko, Kirill Nazarov, Anastasia Gizbrekht, Saleh Alrhmoun, Vera Denisova, Sergey Sennikov

**Affiliations:** 1Laboratory of Molecular Immunology, Research Institute of Fundamental and Clinical Immunology, Novosibirsk 630099, Russia; zavodskii.1448@gmail.com (R.P.-Z.); perik.zavodskaia@gmail.com (O.P.-Z.); shevchenkoja2023@yandex.ru (J.S.); kirill.lacrimator@mail.ru (K.N.); saleh.alrhmoun1@gmail.com (S.A.); 2Clinic of Immunopathology, Research Institute of Fundamental and Clinical Immunology, Novosibirsk 630099, Russia; verden@bk.ru

**Keywords:** erythroid cells, CD45^+^ erythroid cells, antibacterial immunity, lipopolysaccharide, LPS, calprotectin

## Abstract

**Introduction**: Recent studies have shown that Erythroid progenitor cells exhibit a distinct immunosuppressive and immunoregulatory phenotype associated with the response to bacteria. **Methods**: The objective of this study was to comprehensively explore the traits of human bone marrow Erythroid cells through protein–protein interaction network analysis using cytokine secretion analysis, and single-cell immunoproteomic analysis using flow cytometry, as well as the re-analysis of publicly available human and mouse bone marrow Erythroid-cell transcriptomic data. **Results**: Our protein–protein interaction network analysis of human bone marrow Erythroid-cell protein-coding genes identified enrichment in the immune response to lipopolysaccharide, with Calprotectin and Cathepsin G being the main factors. We then mapped the Calprotectin to the CD45^+^ Erythroid cells of both humans and mice via the analysis of the publicly available scRNA-seq data. Additionally, we observed that human bone marrow Erythroid cells secrete cytokines and chemokines, such as IL-1b, IL-8, and IL-18, which are also mainly involved in the immune response to lipopolysaccharide. We also found that human and mouse bone marrow Erythroid-cell conditional media inhibit bacterial growth in vitro. **Discussion**: These findings suggest that both human and mouse bone marrow CD45^+^ Erythroid cells possess the potential to combat pathogenic microbes and thus play a role in innate antimicrobial immunity. **Conclusions**: CD45^+^ Erythroid cells are a potent immunoregulatory cell population in both humans and mice.

## 1. Introduction

Erythropoiesis is a multistep process that begins with pluripotent stem cells and ends with the formation of mature erythrocytes devoid of nuclei [1]. The primary function of erythrocytes is oxygen transport, but immature, nucleus-containing Erythroid cells are also capable of performing other functions, including participation in regulating immune responses [2,3]. Erythroid cells make up a continuum of differentiation called the erythron, which includes: proerythroblasts (Pro Eb), basophilic erythroblasts (Baso Eb), polychromatophilic erythroblasts (Poly Eb), and orthochromatophilic erythroblasts (Ortho Eb) [4,5]. Erythroid cells, being immature cells, are usually absent or very rare in the peripheral blood of healthy adults. Still, they are present in large numbers in cord blood [6,7] and fetal liver [8,9] during human intrauterine development. Erythroid cells can be identified by the presence of surface markers such as CD71 (*TFRC* gene), the transferrin receptor [10], and CD235A (*GYPA* gene), glycophorin A, a major sialoglycoprotein of the Erythroid-cell membrane that is unique to Erythroid cells [11,12]. In addition, Erythroid cells express other surface markers, such as CD36 and CD49d (*ITGA4* gene). The expression of the aforementioned markers decreases with the differentiation of Erythroid cells into mature erythrocytes [9,13,14].

In recent years, it has been suggested that Erythroid cells play a role in immunity. In the context of bacterial infections, it was shown that human Erythroid cells express genes involved in the response to molecules of bacterial origin [9]. This means that there could be a cross-talk between Erythroid cells and the rest of the innate immune system through cytokines or other signaling molecules produced during an infection that is currently neglected.

Despite the significant progress in understanding the pattern of erythrocyte differentiation, the molecular mechanisms involved in the immunomodulatory properties of immature Erythroid cells remain poorly understood. In this work, we decided to perform the systematic protein–protein interaction network analysis of human bone marrow Erythroid-cell protein-coding genes, investigate the cytokine secretome of adult human bone marrow Erythroid cells via Bio-Plex platform, investigate all the subpopulations and stages of Erythroid-cell differentiation by measuring the protein expression of Arginase 1 (*ARG1* gene), CD36, CD45, CD49d, CD71, CD235a, CD184 (*CXCR4* gene), and Galectin 3 (*LGALS3* gene) on single Erythroid cells using flow cytometry, perform a bacterial growth inhibition assay using Erythroid-cell-conditioned media, as well as perform the re-analysis of publicly available human and mouse Erythroid-cell transcriptomic data.

## 2. Materials and Methods

### 2.1. Construction of a Network of Protein–Protein Interactions of the Immune Transcriptome of Human Bone Marrow Erythroid Cells

To evaluate the functional interactions of immune transcriptome genes expressed in Erythroid cells, we used Cytoscape software version 3.10.3 [15] to determine the protein–protein interactions of protein products of genes detected in Erythroid nucleated cells using publicly available data [9,13,14]. To perform the analysis, we combined the lists of found genes from all gene expression experiments, removed singleton genes and genes associated with cell division and metabolism, and imported a new protein–protein interaction network from the STRING database for the remaining connected protein-coding genes [16]. A confidence criterion of 0.8 in the presence of protein–protein interactions was used to create network edges. Next, we performed an enrichment analysis in the Gene Ontology “Biological Process” terms, highlighted all graph nodes included in the biological process “Response to lipopolysaccharide” in yellow, and highlighted all graph nodes included in the extracellular trap formation via a black arc.

### 2.2. Human Bone Marrow Erythroid-Cell Immune Transcriptome Data Re-Analysis

We used CD235a^+^ human bone marrow Erythroid-cell immune transcriptome data (GSE231655) [13] to search for the gene expression of pathogen sensors. We preprocessed and normalized data as described [13], exported the normalized data, manually selected the pathogen sensor genes, log2-transformed the resulting data, and created a heatmap of the selected pathogen sensor genes via bioinfokit [17].

### 2.3. Bone Marrow Sample Collection and Processing

Bone marrow samples were obtained from healthy men and women. The age of the study participants ranged from 25 to 29 years, without any comorbidities or clinical signs of anemia (*n* = 8: *n* = 4—male, *n* = 4—female). We obtained written informed consent from all bone marrow donors participating in the study. This study was conducted in accordance with regional regulatory requirements and the ethical principles outlined in the Declaration of Helsinki.

### 2.4. Bone Marrow Mononuclear Cell Isolation

We collected bone marrow aspirates (2–5 mL) in EDTA-coated tubes. We isolated mononuclear cells from the bone marrow using density gradient centrifugation with Ficoll-Paque (Thermo Fisher Scientific, Waltham, MA, USA) at a density of 1.077 g/mL. The centrifugation was performed at 266 RCF for 30 min, following previously established protocols [9,13,14].

### 2.5. Erythroid-Cell Magnetic Separation

We performed magnetic separation of the bone marrow mononuclear cells using CD235a MicroBeads (Miltenyi Biotec, 130-050-501, Bergisch Gladbach, Cologne, Germany) according to the manufacturer’s protocols as performed previously [9,13].

### 2.6. Erythroid-Cell Culturing

We cultured the Erythroid cells in X-VIVO 10 serum-free medium (Lonza, Basel, Switzerland) with the addition of insulin–transferrin (PanEco, Moscow, Russia) for 24 h at a concentration of 1 million cells per mL of the medium to support their viability at 37 °C and measure the culture medium’s cytokines afterward, as performed previously [13].

### 2.7. Erythroid-Cell Culture-Medium Harvesting

We separated the Erythroid cells’ culture medium from the Erythroid cells after 24 h of culturing. We performed the separation by centrifugation at 1500 rpm for 10 min, transferred the cell culture medium into new 1.5 mL tubes with the addition of BSA up to a total concentration of 0.5%, and froze the cell culture medium at −80 °C until cytokine quantification, as performed previously [13].

### 2.8. Cytokine Quantification in Culture Medium

We used 50 μL of culture medium from bone marrow Erythroid cells (*n* = 8) in duplicate to quantify cytokines using the Bio-Plex Pro Human Cytokine Screening Panel, 48-Plex (BioRad, #12007283, Hercules, CA, USA) on a Bio-Plex 200 instrument (BioRad, Hercules, CA, USA). We log2-transformed the Bio-Plex data and visualized the detected secreted proteins in a heatmap using bioinfokit version 2.1.4 [17]. Additionally, we performed Gene Ontology Biological Process overrepresentation analysis of the detected cytokines using GSEApy 1.1.8 [18].

### 2.9. Murine Bone Marrow Hematopoietic Stem-Cell Atlas Re-Analysis

We used publicly available murine single hematopoietic stem-cell atlas data (“Erythroid_and_monocyte_lineage_adata_no_gaps.h5ad” available at https://milton.cshl.edu/data/HSC_atlas/, accessed on 24 February 2024) (*n* = 9) to search for the “Response to Lipopolysaccharide” pathway genes in murine Erythroid cells. We imported the h5ad file into Seurat V5 [19], converted the file into h5seurat via LoadH5Seurat, imported the h5seuratfile, performed data filtering (*Tfrc* > 0 and *Gypa* > 0, 16,667 cells were retained after data filtering), performed data normalization using SCTransform V2, performed dimensionality reduction using principal component analysis (PCA), performed data integration using Harmony [20], performed UMAP dimensionality reduction using 21 Harmony-corrected PCs, and found neighboring cells and their clusters. We identified Erythroid-cell clusters using *Gypa*, *Tfrc*, *Hba-a2*, and *Ptpr4a2* (CD45) marker expressions. Clusters were merged and renamed to match the stages of Erythroid-cell differentiation. We then created a *DotPlot* and a *FeaturePlot* of the genes with the detected expression in murine Erythroid cells [21,22,23], as well as the “Response to Lipopolysaccharide” genes, such as *S100a8*, *S100a9*, and *Ctsg*. We also created a stacked bar plot of the cluster percentage via ggplot2.

### 2.10. Mouse Erythroid-Cell Immune Transcriptome Data Re-Analysis

We used previously published murine Erythroid-cell immune transcriptome data (bone marrow—normal condition, hemolytic anemia, acute blood loss, and acute hypoxia; spleen—normal condition; and fetal liver—normal condition) [21,22,23] to perform an integrated analysis of murine Erythroid-cell immunome. We performed probe count normalization and sample QC in nSolver 4 using the *Rpl19*, *Gapdh*, *Oaz1*, *Ppia*, and *Eef1g* housekeeping genes included in the panel. We then performed background thresholding on the normalized data to remove non-expressing genes. We determined the background level as the mean of the POS_D controls and removed the genes that were below the background level in all samples, as described previously [21,22,23]. Normalized log2-transformed probe count data are available in the Appendix A. We then used BulkOmicsTools (https://github.com/Perik-Zavodskii/BulkOmicsTools, accessed on 12 March 2024) to perform PCA (principal component analysis) dimensionality reduction and created a heatmap of the murine Erythroid-cell immune transcriptome genes. To test for functional interactions of murine Erythroid-cell immune transcriptome genes, we used Cytoscape software version 3.10.3 [15]. We input a list of genes with the detected expression in all samples, removed singleton genes and genes associated with cell division and metabolism, and imported a new protein–protein interaction network from the STRING database for the remaining connected protein-coding genes [16]. A confidence criterion of 0.8 in the presence of protein–protein interactions was used to create network edges.

### 2.11. Flow Cytometry of Human Bone Marrow Erythroid Cells

A total of 2 × 10^6^ bone marrow mononuclear cells were washed in PBS containing 0.09% NaN_3_ and stained with BioLegend (San Diego, CA, USA) antibodies #125420 APC anti-mouse/human Mac-2 (Galectin 3/*LGALS3* gene), #306522 Brilliant Violet 605™ anti-human CD184 (*CXCR4* gene), #304328 APC/Cyanine7 anti-human CD49d, #334114 PerCP/Cyanine5. 5 anti-human CD71, #349104 FITC anti-human CD235a (Glycophorin A), and #MABF1611 VioletFluor™ 450 Anti-human CD45 antibody (Sigma-Aldrich, St. Louis, MO, USA) according to the manufacturers’ protocols. We then washed the cells after 30 min of incubation at 4 °C in the dark with 0.5 mL of PBS containing 0.09% NaN_3_. We then added #423113 Zombie Violet™ (BioLegend, San Diego, CA, USA) to all samples before fixation. To analyze intracellular and intranuclear proteins, the #424401 True-Nuclear™ buffer set (BioLegend, San Diego, CA, USA) was used to perform fixation and permeabilization. After permeabilization, cells were stained with #369704 PE anti-human Arginase 1 (*ARG1* gene) antibody according to the manufacturer’s protocol. Cells were then washed with 0.5 mL of PBS containing 0.09% NaN_3_ after 30 min of incubation at 4 °C in the dark. Compensation beads #424601 (BioLegend, San Diego, CA, USA) were used to compensate for fluorescence. Attune NxT flow cytometer gating software gated cells from detritus, singletons from cells, live cells from singletons, and finally CD235-positive cells (Erythroid cells) from live cells, and exported them as FCS files.

### 2.12. Human Bone Marrow Erythroid-Cell Flow Cytometry Data Analysis

We converted Fcs files to csv files using fcsparser (https://github.com/eyurtsev/fcsparser, accessed on 24 February 2024), imported CSV files of flow cytometry data into Seurat [19], performed data QC (nCountAb < 1.000.000), performed data normalization using CLR, performed data filtering (CD71 > 0 and CD235a > 0.2, 65,541 cells were retained after data filtering), performed dimensionality reduction using principal component analysis (PCA), performed data integration using Harmony [20], performed UMAP dimensionality reduction using 7 Harmony-corrected PCs, and found neighboring cells and their clusters as described previously [24]. We identified clusters using marker expression, and merged and renamed clusters to match the stages of Erythroid-cell differentiation.

### 2.13. Bacterial Growth Inhibition Assay

We used the conditioned media collected from human and mouse bone marrow Erythroid cells to evaluate their antibacterial activity. Mouse bone marrow Erythroid-cell conditional medium was obtained previously [21]. We performed a bacterial growth inhibition assay using the *E. coli* NebStable strain (New England Biolabs, Ipswich, MA, USA). The bacteria were cultured overnight at 30 °C in LB broth diluted to an initial OD600 of 0.01 and mixed with Erythroid-cell medium (LBB medium 4:1). We used LB broth with bacteria as a positive control, LB with bacteria supplemented with 50 μg/mL kanamycin as a negative control, and LB without bacteria as a blank. The assay was performed in duplicate. The absorbance at 595 nm was measured using a Varioskan Lux plate reader (Thermo Fisher Scientific, Waltham, MA, USA) every 15 min for 18 h. The plate was maintained at 30 °C and shaken continuously at 60 rpm throughout the experiment.

## 3. Results

### 3.1. Protein-Coding Genes Expressed by the Human Bone Marrow Erythroid Cells Form a Connected Net of Protein–Protein Interactions Involved in the Immune Response to LPS

In this work, in order to systematically assess the human bone marrow Erythroid-cell immunome, we performed a protein–protein interaction network analysis using publicly available data via Cytoscape and STRING. We found the presence of a functional network of protein–protein interactions that were included in the “Response to lipopolysaccharide” Gene Ontology Biological Process Term (Figure 1A, Table 1). In addition, we found a subnetwork of the aforementioned network that contained genes responsible for the formation of extracellular neutrophil traps—Calprotectin (S100A8, S100A9), Cathepsin G (*CTSG* gene), and Alpha-3 defensin (*DEFA3* gene), and thus for directly combating bacteria. We also identified other direct regulators of antibacterial inflammatory response, such as IL-1b, IL-8, and IL-18 cytokines and chemokines (Figure 1A).

As we found the “Response to Lipopolysaccharide” to be the main gene expression signature in the human bone marrow Erythroid cells, our next step was to assess the means by which human bone marrow Erythroid cells could recognize LPS-bearing bacteria using the publicly available human bone marrow Erythroid-cell immune transcriptome data. We found that Complement receptor 1 (*CR1* gene) was the only pathogen recognition-related gene expressed by human bone marrow Erythroid cells that also had the same gene expression levels as the genes involved in the extracellular trap formation—*S100A8*, *S100A9*, and *CTSG* genes (Figure 1B).

### 3.2. Human Bone Marrow Erythroid Cells Secrete Cytokines Involved in the Immune Response to LPS

Among the genes that formed the “Response to lipopolysaccharide” gene expression signature were cytokines and chemokines, such as IL-1b, IL-8, and IL-18, which play a great role in the antibacterial inflammatory response. To check for the factual presence of the aforementioned cytokine production and secretion, we then performed a bulk secretomic study of normal adult bone marrow Erythroid-cell-conditioned media. Our analysis showed that there is indeed IL-1b, IL-8, and IL-18 cytokine and chemokine secretion, as well as MIF chemokine secretion (Figure 2A). These detected cytokines and chemokines were enriched in the “Response to Lipopolysaccharide” and the “Lipopolysaccharide-Mediated Signaling Pathway” Gene Ontology Biological Process terms (Figure 2B), thus confirming our findings that the “Response to Lipopolysaccharide” signature is the dominant among human bone marrow bone marrow Erythroid cells, even on the secretory level.

### 3.3. Response to LPS Pathway Genes Is Expressed by the CD45^+^ Murine Bone Marrow Erythroid Cells

Next, we decided to use the publicly available single hematopoietic stem-cell (HSC) atlas data to check for the presence of the “Response to Lipopolysaccharide” gene expression signature in murine bone marrow Erythroid cells (Figure 3).

We observed all stages of murine bone marrow Erythroid-cell stages of differentiation—proerythroblasts (Pro Eb), basophilic erythroblasts (Baso Eb), polychromatophilic erythroblasts (Poly Eb), and orthochromatophilic erythroblasts (Ortho Eb). We also observed the *Ptp4a2*/CD45^+^ erythroblasts (CD45^+^ Eb) (Figure 3B,C).

We found that the “Response to Lipopolysaccharide” gene expression signature (*S100a8*, *S100a9*, and *Ctsg* genes, as well as the *Camp* gene, which was exclusive to murine Erythroid cells) was mainly found in the CD45^+^ Eb, with some also found in the Pro Eb. Murine CD45^+^ Eb also contained most of the *Tgfb1* gene, the expression of which is absent in the human bone marrow Erythroid cells [13,14] (Figure 3D). CD45^+^ Eb comprised 14.5% of murine bone marrow Erythroid cells (Figure 3A).

On the other hand, we observed universal *Lgals9* and *Mif* gene expression in the murine bone marrow Erythroid cells but not any Arginase (*Arg1* and *Arg2* genes) or PD-L1 (*Cd274* gene) gene expression (Figure 3D).

We then decided to perform an integrated immunome analysis of murine Erythroid cells using the publicly available immune transcriptome data—normal bone marrow (Eb BM N), normal spleen (Eb Spl N), and normal fetal liver (Eb FL N) murine Erythroid cells, as well as the bone marrow post-acute hypoxia (Eb BM hyp02), bone marrow post-hemolytic anemia (Eb BM HA), and bone marrow post-acute blood loss (Eb BM ABL) murine Erythroid cells.

First, we performed PCA on the samples and observed that Erythroid cells from all three tissues clustered separately (Figure 4A). Upon the inspection of their immune transcriptome genes, we observed that BM Erythroid cells had a high expression of antimicrobial response genes (*S100a8*, *S100a9*, *Ctsg*, *Clecl4e*, *Clec5a*, and *Camp* genes) and that the expression of such genes is increased following any acute hemopoiesis-disturbing event (acute blood loss, acute hypoxia, acute hemolytic anemia).

Fetal liver Erythroid cells displayed high levels of *Mx1* and *Ifna1* antiviral response genes, as well as high *Mif* chemokine gene expression.

Splenic Erythroid cells were the most interesting of the three as they had high expression levels of MHC Class II antigen-presentation pathway genes, as well as a high expression of immunosuppressive *Cd274* (PD-L1) and *Tgfb1* genes (Figure 4B).

We then performed a protein–protein interaction network analysis of murine Erythroid-cell immunome and observed the presence of the extracellular trap (ET) formation complex in murine Erythroid cells as well as in human Erythroid cells (Figure 4C).

The observed differences in the gene expression levels between normal bone marrow and normal spleen Erythroid cells suggest a tissue-driven polarization into antimicrobial Erythroid cells (bone marrow) and antigen-presenting tolerogenic Erythroid cells (spleen) (Figure 4D).

### 3.4. CD45^+^ Human Bone Marrow Erythroid-Cell Proteome Maps to a Calprotectin-Positive Cell Population Found in the Human Bone Marrow Erythroid-Cell scRNA-Seq Data

Next, we conducted a single-cell proteomic study of the human bone marrow Erythroid cells to find distinct Erythroid-cell stages of differentiation and assess the studied protein–protein co-expression in them.

To cover all known human bone marrow Erythroid-cell clusters, we studied CD235a, CD71 (common human Erythroid-cell markers), CD36, CD49d (Erythroid-cell differentiation markers), CXCR4, Galectin 3 (Calprotectin-expressing Erythroid-cell positive and negative markers, respectively [13]), and Arginase 1 (Arginase 1-positive Orthochromatic erythroblast selective marker [13,14]) protein expression. We also studied the CD45 protein expression on human bone marrow Erythroid cells due to its expression on the murine Erythroid cells that expressed the “Response to LPS” signature genes (Figure 5A,C,D).

In our analysis, we were able to identify all predicted Erythroid-cell stages of differentiation and subpopulations using clustering: Pro Eb, Baso Eb, Poly Eb, and Orthro Eb, as well as the Arginase 1-positive Orthochromatic erythroblasts (ARG1^+^ Ortho Eb) and CD45^+^ Erythroid cells (CD45^+^ Eb) (Figure 5B,C). We also validated the data via non-clustering approaches, such as Spearman correlation and pseudotime trajectory inference (see Appendix A).

CD45^+^ Erythroid cells exhibited the unique expression of the chemokine receptor CXCR4 (CD184) and the absence of Galectin 3 expression among human bone marrow Erythroid cells, which allowed us to map the CD45 protein expression to the *S100A9*-expressing human bone marrow Erythroid cells, the same as in murine bone marrow Erythroid cells (Figure 5C, Appendix A).

Arginase 1-positive Orthochromatic erythroblasts comprised 20.64 ± 5.61% of human bone marrow Erythroid cells, and CD45^+^ Erythroid cells comprised 2.19 ± 1.05% of human bone marrow Erythroid cells.

### 3.5. Human and Mouse Bone Marrow Erythroid-Cell-Conditioned Media Inhibit Bacterial Growth In Vitro

We then tested the antibacterial activity of the conditioned medium with human bone marrow Erythroid cells in vitro for 18 h every 15 min (1 part medium, 4 parts LBB) and observed complete inhibition of *E. coli* growth in vitro (Figure 6).

## 4. Discussion

In this study, we conducted a systematic study of the immunome of human and mouse bone marrow Erythroid cells.

Our cytokine production assay and protein–protein interaction network analysis of human bone marrow Erythroid cells revealed enrichment in the LPS response through IL-1b, IL-8, IL-18, MIF, Calprotectin, Alpha-3 defensin, and Cathepsin G.

Calprotectin and Alpha-3 defensin are known antimicrobial proteins with a broad spectrum of specificity [25,26,27,28,29] that were found to be restricted to two populations of Erythroid cells—Myeloid-like and DEFA3^+^ Erythroid cells [13]. In this study, we were able to map the Myeloid-like Erythroid-cell transcriptomic profile to the proteomic profile of CD45^+^ Erythroid cells, thus allowing the identification of Myeloid-like Erythroid cells as CD235a^+^ CD45^+^ Erythroid cells. It is also known that Myeloid-like Erythroid cells have *CXCL8* (IL-8) [13]. As we have observed that Erythroid cells indeed produce IL-8, we are now able to map its production to CD235a^+^ CD45^+^ Erythroid cells as well.

Our re-analysis of publicly available Erythroid-cell transcriptomic data allowed us to single out the molecule that could be responsible for microbial recognition in human bone marrow Erythroid cells—Complement Receptor 1 (*CR1* gene). We did not detect any TLR4 gene expression that recognized LPS [30,31], which suggests that the “Response to Lipopolysaccharide” gene signature is constitutively expressed and does not require LPS itself for its induction. Normally, Complement Receptor 1 allows red blood cells to capture and eliminate complement-antibody–antigen complexes from the bloodstream [32,33], which suggests that Erythroid cells can also recognize such complexes, as they have prominent *CR1* gene expression.

We also observed that Calprotectin gene expression (*S100A8* and *S100A9* genes) is on the same level as the Cathepsin G (*CTSG* gene) gene expression by re-analyzing the human bone marrow Erythroid-cell transcriptomic data. This suggests that CD45^+^ Erythroid cells that express Calprotectin can also express Cathepsin G. In fact, we observed such co-expression in CD45^+^ murine bone marrow Erythroid cells by re-analyzing publicly available scRNA-seq data, which means that CD45^+^ bone marrow Erythroid cells in both humans and mice could be involved in the antimicrobial response. Calprotectin and Cathepsin G are also known to form extracellular traps in neutrophils [34,35,36], which could mean that CD45^+^ bone marrow Erythroid cells in both humans and mice are capable of forming extracellular traps to combat microbial pathogens. As we observed CD45-positive Erythroid cells to be similar in their transcriptomic profiles among humans and mice, our finding might be transferred to other mammals as well.

Both transcriptomic [13] and proteomic analyses of the human bone marrow CD45^+^ Erythroid cells revealed that these cells exhibit high CXCR4 (CD184) chemokine receptor expression. CXCR4 is responsible for the migration toward its ligand, CXCL12. This makes CD45^+^ Erythroid cells migration-ready cells, which are probably restricted to the bone marrow, as CXCL12 is mainly secreted by bone marrow stromal cells [37,38,39], but could potentially migrate elsewhere in case of extramedullary erythropoiesis. We depicted all of the aforementioned assumptions on the human bone marrow CD45^+^ Erythroid cells in Figure 7.

Human CD45^+^ Erythroid cells also completely lacked the expression of Galectin 3, a surface immunosuppressive protein crucial for T-cell immunoregulation [40,41,42]. This suggests that CD45^+^ Erythroid cells cannot perform Galectin 3-based immunosuppression of T cells.

Our proteomic analysis of single human bone marrow Erythroid cells also showed the presence of a distinct immunosuppressive subpopulation among normal bone marrow mononuclear cells, even in healthy donors, Arginase 1-positive Erythroid cells, thus revealing the presence of basal immunosuppression in the bone marrow. The Arginase 1-positive Orthochromatic erythroblasts might hold significant importance in evaluating different immunosuppressive microenvironments aside from the normal bone marrow microenvironment, such as the extramedullary erythropoiesis microenvironment or the tumor microenvironment.

We also observed that human and mouse bone marrow Erythroid-cell-conditioned media fully suppressed *E. coli* growth in vitro, which can be explained by the presence of either Calprotectin or Alpha-defensin 3 in the conditioned media. These findings also support the performed bioinformatic analysis and further emphasize the antimicrobial potential of bone marrow Erythroid cells.

Our re-analysis of murine Erythroid-cell immune transcriptome data also revealed that splenic murine Erythroid cells express MHC Class II antigen-presentation genes, as well as *Fcgr2b*, *Mif*, *Cd274* (PD-L1), and *Tgfb1*, and have no expression of *Cd80*/*Cd86* genes. This, together with their co-localization with the CD4 T cells in the spleen yields a perfect ground for the generation of CD4 Treg T cells—splenic Erythroid cells can uptake the IgG-opsonized circulating antigens with their Fc-receptors (*Fcgr2b*), process and present the antigens in an MHC Class II complex, attract CD4 T cells via MIF [43,44], and secrete TGF-b1 in close proximity to CD4 T cells, thus re-programming them into CD4 Treg with the help of the lacking CD80/86 co-stimuli [45,46,47,48,49,50].

While our study provides significant insights into the antimicrobial role of human and mouse bone marrow Erythroid cells, several limitations should be considered. First, this study was based on a relatively small sample size (*n* = 6–8) of human bone marrow donors, all of whom were healthy, young adults. This limited diversity may not fully capture the potential variations in Erythroid-cell function across different age groups, populations, or individuals with underlying health conditions. Additionally, although our in vitro assays demonstrated antimicrobial activity of Erythroid-cell-conditioned media, these results were based on *E. coli*, a model bacterium, and we also did not use either calprotectin or alpha-3 defensin as positive controls. The antimicrobial activity of Erythroid cells against other pathogens, such as viruses or fungi, remains unexplored and could differ depending on the pathogen type.

## 5. Conclusions

Our study provides compelling evidence that both human and mouse bone marrow Erythroid cells possess intrinsic antimicrobial capabilities, challenging the traditional view of erythrocytes as passive oxygen carriers and thus highlighting their potential participation in the innate immune system. These insights not only broaden our understanding of Erythroid-cell functions but also open new avenues for exploring their therapeutic potential in combating infections.

## Figures and Tables

**Figure 1 biomedicines-13-01218-f001:**
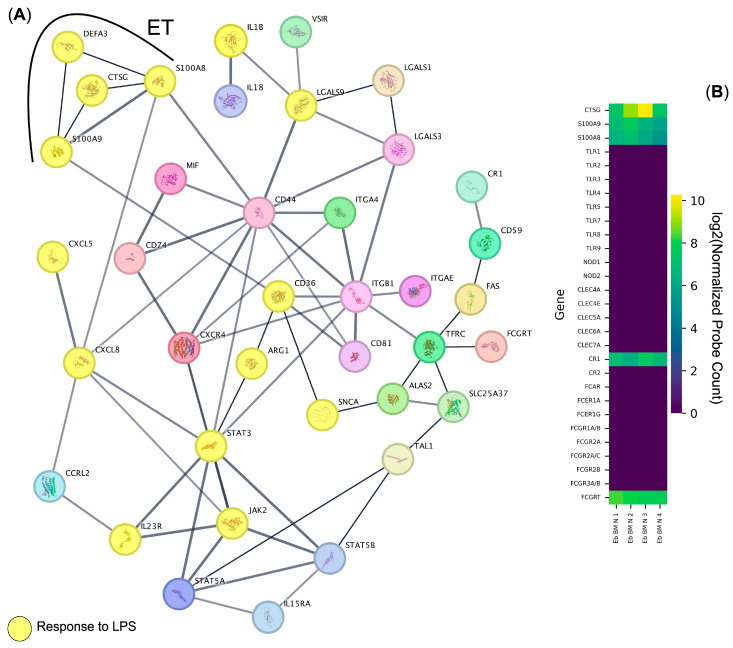
Systematic analysis of the human bone marrow Erythroid-cell immunome. (**A**) Protein–protein interaction network analysis of the immune transcriptome of Erythroid nucleated cells from the bone marrow of healthy adult donors. Genes included in the term “Response to Lipopolysaccharide” of the Gene Ontology “Biological Process” database are depicted as bubbles with their protein structure inside and are highlighted in yellow, and the genes included in the extracellular trap (ET) formation are highlighted via a black arc. (**B**) Heat map of the pathogen sensor and the extracellular trap formation genes expressed by normal human bone marrow Erythroid cells (Eb BM N) (*n* = 4).

**Figure 2 biomedicines-13-01218-f002:**
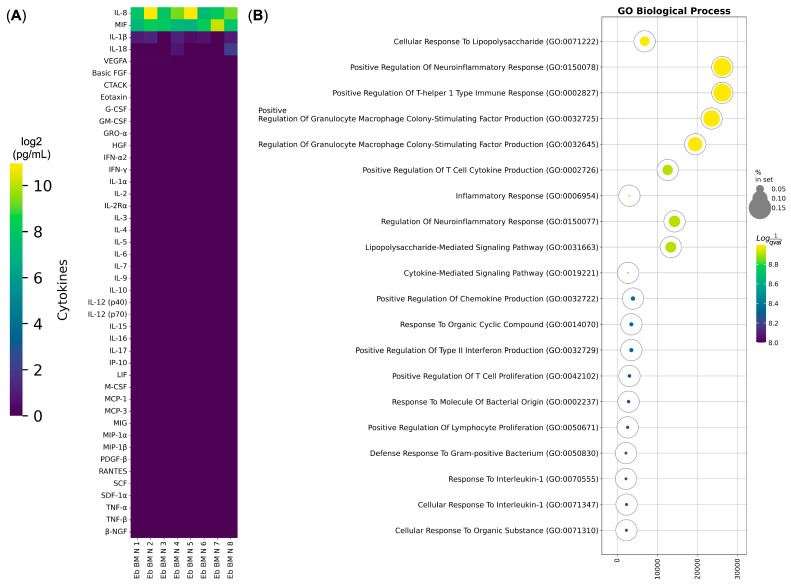
Cytokine secretion analysis of human bone marrow Erythroid-cell-conditioned media via Bio-Plex. (**A**) Heatmap of cytokines secreted by normal human bone marrow Erythroid cells (Eb BM N). The heatmap displays log2-transformed values of cytokine concentrations in pg/mL, where yellow indicates the highest detected secretion levels and deep purple represents the absence of cytokine secretion. (**B**) Gene Ontology Biological Process overrepresentation analysis of the cytokines detected in human adult bone marrow Erythroid cells. The yellow color corresponds to the lowest *q*-value, while deep purple represents the highest *q*-value. The size of each bubble reflects the percentage of detected proteins included in the Gene Ontology Biological Process analysis relative to the full set in the database.

**Figure 3 biomedicines-13-01218-f003:**
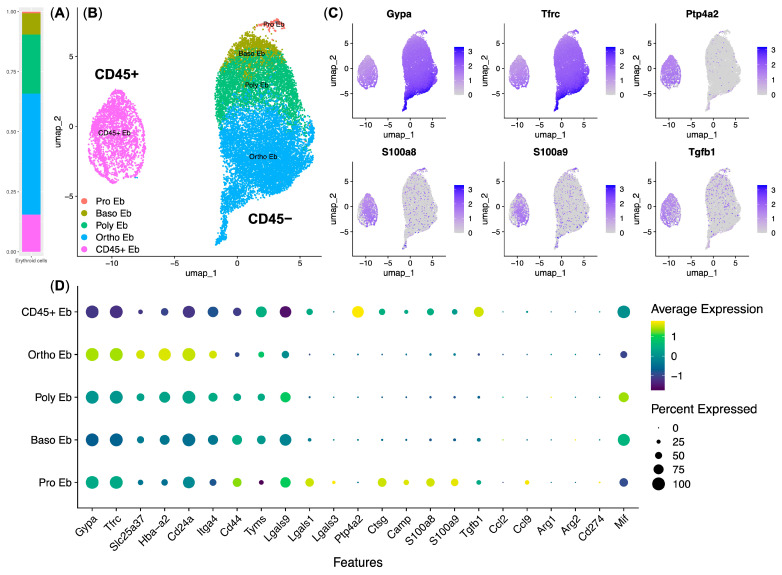
Re-analysis of the murine bone marrow Erythroid cells using single HSC atlas data (*n* = 9, 16.667 single cells). (**A**) Stacked bar plot of the cell cluster percentages, (**B**) UMAP plot of Erythroid-cell clusters, (**C**) feature plots of the key murine Erythroid cells and responses to LPS genes, and (**D**) dot plot of the key murine Erythroid-cell and immune transcriptome genes.

**Figure 4 biomedicines-13-01218-f004:**
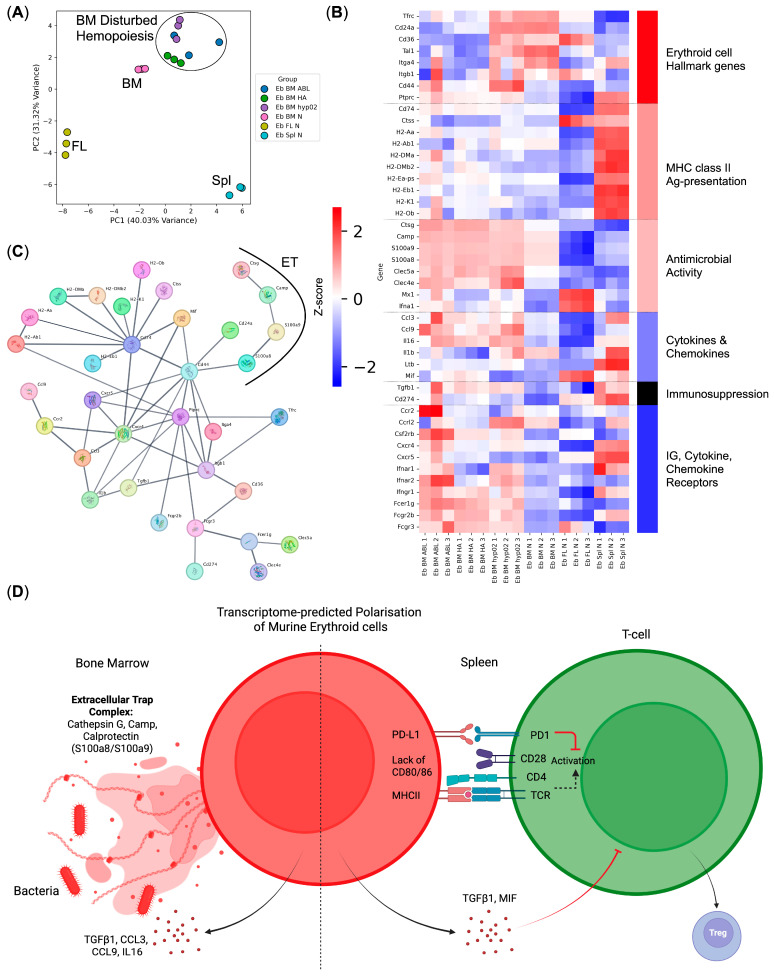
Integrative analysis of murine Erythroid-cell immunome. (**A**) PCA plot of the samples, BM—bone marrow, Spl—spleen, and FL—fetal liver; (**B**) heat map of the murine Erythroid-cell immune transcriptome genes; (**C**) protein–protein interaction network of the Erythroid-cell immune transcriptome protein-coding genes (extracellular trap (ET) complex genes are depicted as bubbles with their protein structure inside and are highlighted via a black arc); (**D**) transcriptome-predicted polarization of murine bone marrow and spleen Erythroid cells—bone marrow murine Erythroid cells express genes involved in antibacterial immunity, while splenic murine Erythroid cells express genes involved in the exogenous antigen presentation via MHC class II.

**Figure 5 biomedicines-13-01218-f005:**
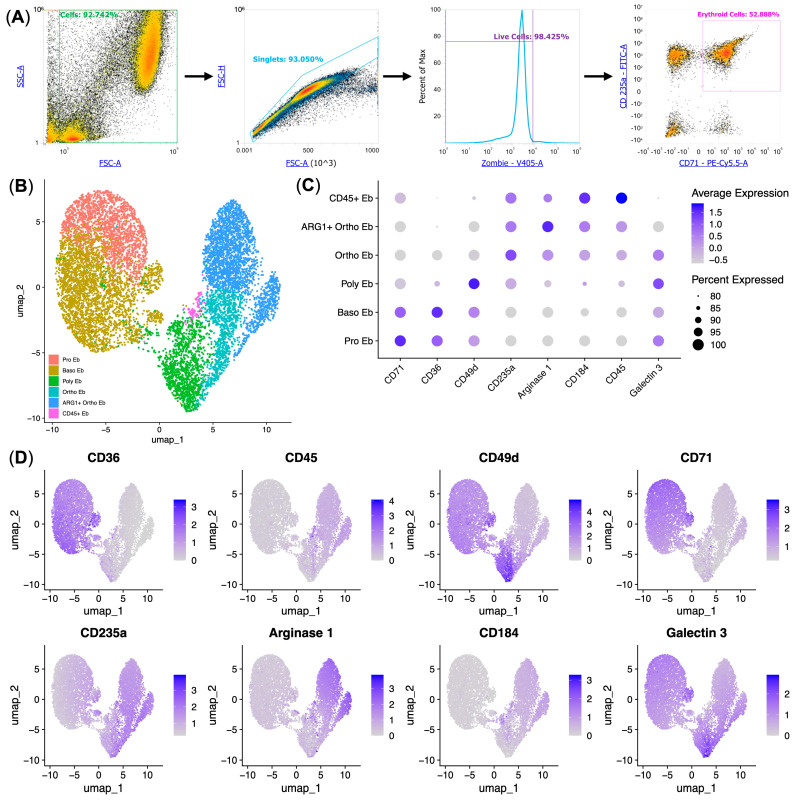
Analysis of protein expression in single Erythroid cells by flow cytometry (*n* = 8, 65.541 single cells). (**A**) Schematic of Erythroid-cell gating using density plots and a histogram; (**B**) UMAP plot of Erythroid-cell clusters; (**C**) dot plot of marker expression per cluster, protein expression values are standardized, and the size of the dots represents the percentage of cells positive for the marker; (**D**) UMAP feature plots of marker expression.

**Figure 6 biomedicines-13-01218-f006:**
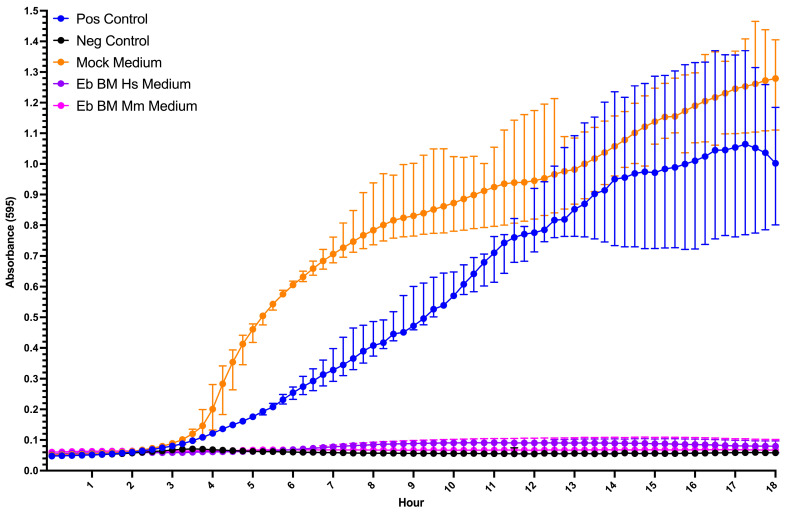
In vitro bacterial growth inhibition assay: positive (Pos) control—*E. coli* in LBB, negative (Neg) control—*E. coli* in LBB with kanamycin, mock medium—*E. coli* in X-VIVO 10 medium 1:4 LBB, Eb BM Hs medium—*E. coli* in human bone marrow Erythroid-cell-conditional medium 1:4 LBB, and Eb BM Mm medium—*E. coli* in mouse bone marrow Erythroid-cell-conditional medium 1:4 LBB (*n* = 6).

**Figure 7 biomedicines-13-01218-f007:**
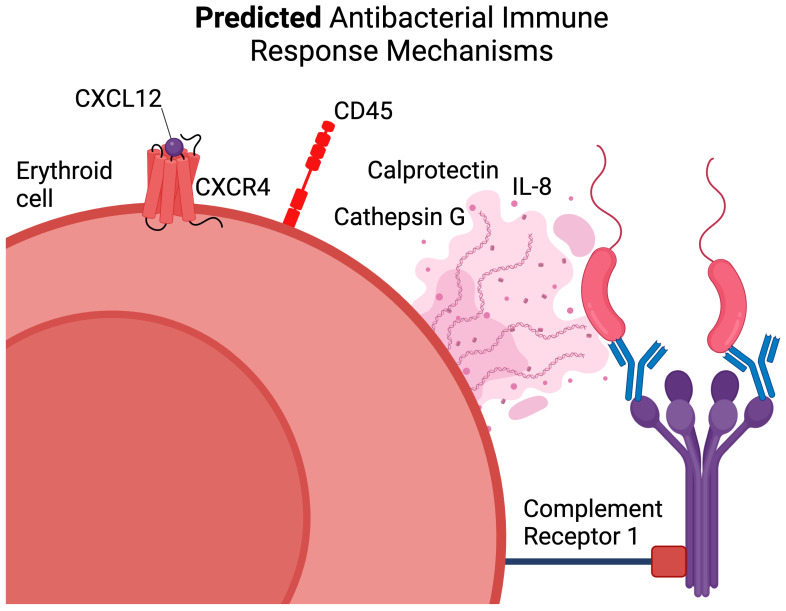
Overview of the predicted mechanisms by which CD45^+^ human bone marrow Erythroid cells can facilitate antimicrobial response. Erythroid cells express the chemokine receptor CXCR4, which interacts with CXCL12 to facilitate cell migration, and secrete antimicrobial proteins such as Calprotectin and Cathepsin G, which are involved in the defense against pathogens. Additionally, CD45^+^ Erythroid cells produce IL-8, a chemokine involved in the recruitment of immune cells to sites of infection. Complement Receptor 1 (CR1) on the CD45^+^ Erythroid cells is shown as a potential receptor for recognizing and binding complement-opsonized pathogens, contributing to the CD45^+^ Erythroid-cell antimicrobial function.

**Table 1 biomedicines-13-01218-t001:** Gene Ontology Biological Process overrepresentation analysis of the protein-coding human adult bone marrow Erythroid-cell genes with detected expression that are also involved in protein–protein interactions with each other.

Gene Ontology Biological Process Term	*q*-Value	Hits	Out Of	Score	Genes
Response to lipopolysaccharide	0.000000001	14	314	446	*CTSG*, *CXCL5*, *CXCL8*, *STAT3*, *IL1B*, *IL23R*, *DEFA3*, *ARG1*, *S100A8*, *S100A9*, *JAK2*, *LGALS9*, *CD36*, *SNCA*
Immune cell migration	0.000000001	11	249	442	*CTSG*, *LGALS3*, *STAT5B*, *CXCL5*, *CXCL8*, *S100A8*, *S100A12*, *S100A9*, *ITGB1*, *ITGA4*, *CXCR4*
Regulation of intercellular adhesion	0.000000012	15	368	408	*CD74*, *LGALS1*, *CTSG*, *LGALS3*, *CD81*, *STAT5B*, *IL23R*, *ARG1*, *CR1*, *JAK2*, *TFRC*, *VSIR*, *LGALS9*, *ITGA4*, *CD44*
Positive regulation of the immune effector process	0.000000038	10	264	379	*CD74*, *MIF*, *CD81*, *STAT5B*, *IL23R*, *ARG1*, *CR1*, *TFRC*, *LGALS9*, *CD36*
Humoral immune response	0.000000697	9	268	336	*CTSG*, *LGALS3*, *CD81*, *CXCL5*, *CXCL8*, *DEFA3*, *CR1*, *S100A12*, *S100A9*
Positive regulation of the response to an external stimulus	0.000000732	14	453	309	*CD74*, *MIF*, *LGALS1*, *CD81*, *STAT5B*, *CXCL8*, *ARG1*, *S100A8*, *S100A12*, *S100A9*, *JAK2*, *LGALS9*, *CXCR4*, *SNCA*
Activation of immune cells	0.000001202	14	574	244	*CD74*, *LGALS1*, *CTSG*, *CD81*, *STAT3*, *STAT5B*, *CXCL8*, *S100A12*, *JAK2*, *ITGB1*, *ITGA4*, *IL15RA*, *CD44*, *SNCA*
Positive regulation of immune cell migration	0.000001130	11	529	208	*CD74*, *LGALS3*, *STAT3*, *CXCL8*, *STAT5A*, *JAK2*, *VSIR*, *LGALS9*, *ITGB1*, *ITGA4*, *CXCR4*
Regulation of the apoptotic process	0.000009600	15	1462	103	*CD74*, *MIF*, *LGALS1*, *LGALS3*, *STAT5B*, *S100A8*, *S100A9*, *JAK2*, *TFRC*, *LGALS9*, *ITGB1*, *ITGA4*, *CD44*, *FAS*, *SNCA*

## Data Availability

The data obtained in this manuscript and the code used in this manuscript are available on Zenodo: https://doi.org/10.5281/zenodo.14513371, accessed on 5 May 2025.

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
