# Peer review of "Human and Mouse Bone Marrow CD45+ Erythroid Cells Have a Constitutive Expression of Antibacterial Immune Response Signature Genes"

_biomedicines, 2025, doi:10.3390/biomedicines13051218_

Round 1

Reviewer 1 Report

Comments and Suggestions for Authors

The authors examined signature genes implicated in the antimicrobial immune response of erythroblastic cells. This study showed that erythroblastic cells' production of calprotectin and alpha-defensin 3 inhibited the proliferation of Escherichia coli. The reviewer recommends conducting further studies to validate this explanation.

Major comments

Figure 6 illustrates that the culture supernatant of human bone marrow erythroblast cells entirely suppresses the growth of E. coli; however, to validate this, the reviewer recommends verifying whether calprotectin and alpha-defensin 3 block the growth of E. coli.

Moreover, are the erythroblast cells themselves not implicated in antimicrobial activity?

Minor comments

Line 134: Is the period in “16.667 cells” an error for a comma (16,667 cells is correct)?

Line 182: Is "Figure 2A" suitable?

Is CXCL-8 (IL-8) exactly classified as a chemokine?

Lines 249 and 251: Are Figures 3A and 3B correct, respectively?

Please provide a more comprehensive elucidation for the legend accompanying Figure 4D.

Please include a concise explanation of the procedure in the legend for Figure 7.

Please provide a summary of the conclusions of this paper in a distinct section.

Author Response

Dear Reviewer, Thank You for Your inquiries and observations. In response to the points raised:

Figure 6 illustrates that the culture supernatant of human bone marrow erythroblast cells entirely suppresses the growth of E. coli; however, to validate this, the reviewer recommends verifying whether calprotectin and alpha-defensin 3 block the growth of E. coli.

We have screened published literature to select E. coli as a target susceptible to both Alpha-Defensin 3 and Calprotectin [https://doi.org/10.1039/c8mt00133b, https://doi.org/10.1172/jci112120, https://doi.org/10.1007/s00109-005-0657-1] to perform this assay correctly.  

Moreover, are the erythroblast cells themselves not implicated in antimicrobial activity?

They might be directly involved, yet future more complex studies might be required to asses the direct role of Erythroid cells in the innate immunity.

Line 134: Is the period in “16.667 cells” an error for a comma (16,667 cells is correct)?

We fixed that. Thank You!

Line 182: Is "Figure 2A" suitable?

We fixed this reference!

Is CXCL-8 (IL-8) exactly classified as a chemokine?

Yes, IL-8 is a well-established chemokine [https://doi.org/10.1016/j.pharmthera.2020.107692, https://doi.org/10.1016/j.oftal.2019.11.014, https://doi.org/10.1038/s41423-023-00974-6].

Lines 249 and 251: Are Figures 3A and 3B correct, respectively?

We fixed these references!

Please provide a more comprehensive elucidation for the legend accompanying Figure 4D.

We have added the following explanation: “Transcriptome-predicted polarization of murine bone marrow and spleen Erythroid cells – bone marrow murine Erythroid cells express genes involved in the antibacterial immunity, while splenic murine Erythroid cells express genes involved in the exogenous antigen presentation via MHC class II.”

Please include a concise explanation of the procedure in the legend for Figure 7.

We have revised the legend for Figure 7:

"Figure 7. Overview of the predicted mechanisms by which CD45+ human bone marrow Erythroid cells could facilitate antimicrobial response. Erythroid cells express the chemokine receptor CXCR4, which interacts with CXCL12 to facilitate cell migration, and secrete antimicrobial proteins such as Calprotectin and Cathepsin G, which are involved in the defense against pathogens. Additionally, CD45+ Erythroid cells produce IL-8, a cytokine involved in the recruitment of immune cells to sites of infection. Complement Receptor 1 (CR1) on the CD45+ Erythroid cells is shown as a potential receptor for recognizing and binding complement-opsonized pathogens, contributing to the CD45+ Erythroid cell antimicrobial function."

Please provide a summary of the conclusions of this paper in a distinct section.

We have added a dedicated Conclusions section to the manuscript to succinctly summarize the key findings and their implications:

“Our study provides compelling evidence that both human and mouse bone marrow erythroid cells possess intrinsic antimicrobial capabilities, challenging the traditional view of erythrocytes as passive oxygen carriers and thus highlighting their potential participation in the innate immune system. These insights not only broaden our understanding of erythroid cell functions but also open new avenues for exploring their therapeutic potential in combating infections.“

Reviewer 2 Report

Comments and Suggestions for Authors

In this current article “Human and mouse bone marrow CD45+ Erythroid cells have a 2 constitutive expression of antibacterial immune response 3 signature genes” author has tried to explain human and mouse bone marrow CD45+ Erythroid cells possess the potential to combat pathogenic microbes and thus play a role in innate antimicrobial immunity by using array of methods it can be published by addressing following points

1) Method section need little refinement, specific temperatures in the methods should be mentioned.

2)Conclusion lacks the gravity of the study it can be more polished  and made more effective by adding the direct relevance of the study.

3)Authors need to mention the limitations of the study

4) Will the study be relevant keeping all kinds of variation in species and diversity of regions.

Author Response

Dear Reviewer, Thank You for Your inquiries and observations. In response to the points raised:

1) Method section need little refinement, specific temperatures in the methods should be mentioned.

We appreciate this important suggestion. We have revised the Materials and Methods section to specify the temperature conditions for all relevant procedures. For example, we now state that bone marrow mononuclear cells were isolated by Ficoll density centrifugation at room temperature, that erythroid cells were cultured at 37 °C, and that antibody staining incubations for flow cytometry were performed on ice in the dark. These additions ensure that all experimental protocols are described with the appropriate temperature details, improving the clarity and reproducibility of our methods.

2) Conclusion lacks the gravity of the study it can be more polished  and made more effective by adding the direct relevance of the study.

We agree that the conclusion needs to more strongly convey the importance of our findings. We have added the Conclusion section, which contains the following paragraph:
“Our study provides compelling evidence that both human and mouse bone marrow erythroid cells possess intrinsic antimicrobial capabilities, challenging the traditional view of erythrocytes as passive oxygen carriers and thus highlighting their potential participation in the innate immune system. These insights not only broaden our understanding of erythroid cell functions but also open new avenues for exploring their therapeutic potential in combating infections.“

3)Authors need to mention the limitations of the study

We appreciate the opportunity to clarify the limitations of our study. In response, we have added a dedicated Limitations subsection to the discussion. This new paragraph is:

“While our study provides significant insights into the antimicrobial role of human and mouse bone marrow erythroid cells, several limitations should be considered. First, the study was based on a relatively small sample size (n = 8) of human bone marrow donors, all of whom were healthy young adults. This limited diversity may not fully capture the potential variations in erythroid cell function across different age groups, populations, or individuals with underlying health conditions. Additionally, although our in vitro assays demonstrated antimicrobial activity of erythroid cell-conditioned media, these results were based on E. coli, a model bacterium. The antimicrobial activity of erythroid cells against other pathogens, such as viruses or fungi, remains unexplored and could differ depending on the pathogen type.”

4) Will the study be relevant keeping all kinds of variation in species and diversity of regions.

Indeed, as we have observed CD45-positive Erythroid cells to be stable in their transcriptome among humans and mice, our finding may be transferred to other mammals as well. We have added the paragraph addressing this in the Discussion section.

Round 2

Reviewer 1 Report

Comments and Suggestions for Authors

The authors have mostly responded to the reviewer's comments satisfactorily. Nevertheless, certain concerns persist.

  1. Concerning Figure 6, it is recommended that the authors provide evidence that human bone marrow erythroblast cell culture supernatant entirely suppresses E. coli proliferation by doing confirmatory tests that illustrate the inhibition of E. coli growth upon treatment with either calprotectin or alpha-defensin 3. The reviewer requests that the authors augment the material with supplementary experimental results.
  2. The reviewer recommends that IL-8 be distinctly categorized and characterized as a chemokine in the main text, as it is presently referred to as a cytokine.

Author Response

Dear Reviewer, we currently do not have access to either pure calprotectin or alpha-3 defensin, but we would consider using it as a positive control in our further studies. As of this moment, we have added the lack of either calprotectin or alpha-3 defensin positive controls as a limitation of our study. As for IL-8, we have changed its description to chemokine throughout the manuscript.